# Transforming a Quadruped into a Guide Robot for the Visually Impaired: Formalizing Wayfinding, Interaction Modeling, and Safety Mechanism

**J. Taery Kim**
Georgia Institute of Technology
taerykim@gatech.edu

**Wenhao Yu**
Google DeepMind
magicmelon@deepmind.com

**Yash Kothari**
Georgia Institute of Technology
ykothari3@gatech.edu

**Bruce N. Walker**
Georgia Institute of Technology
bruce.walker@psych.gatech.edu

**Jie Tan**
Google DeepMind
jietan@deepmind.com

**Greg Turk**
Georgia Institute of Technology
turk@cc.gatech.edu

**Sehoon Ha**
Georgia Institute of Technology
sehoonha@gatech.edu

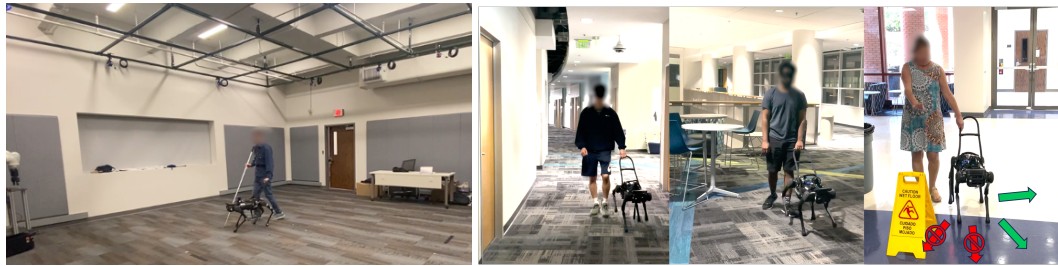

Figure 1: We develop a robot guide dog by formalizing the wayfinding task, developing an interaction model from the collected data (**Left**), and employing action shielding, to guide a user (**Right**).

**Abstract:** This paper explores the principles for transforming a quadrupedal robot into a guide robot for individuals with visual impairments. A guide robot has great potential to resolve the limited availability of guide animals that are accessible to only two to three percent of the potential blind or visually impaired (BVI) users. To build a successful guide robot, our paper explores three key topics: (1) formalizing the navigation mechanism of a guide dog and a human, (2) developing a data-driven model of their interaction, and (3) improving user safety. First, we formalize the wayfinding task of the human-guide robot team using Markov Decision Processes based on the literature and interviews. Then we collect real human-robot interaction data from three visually impaired and six sighted people and develop an interaction model called the "Delayed Harness" to effectively simulate the navigation behaviors of the team. Additionally, we introduce an action shielding mechanism to enhance user safety by predicting and filtering out dangerous actions. We evaluate the developed interaction model and the safety mechanism in simulation, which greatly reduce the prediction errors and the number of collisions, respectively. We also demonstrate the integrated system on a quadrupedal robot with a rigid harness, by guiding users over 100+ m trajectories.

**Keywords:** Assistive Robot, Autonomous Navigation, Interaction Modeling

7th Conference on Robot Learning (CoRL 2023), Atlanta, USA.

# 1  Introduction

Guide dogs play a crucial role in the lives of blind or visually impaired individuals (BVIs) by providing them with increased mobility and independence [1]. Unfortunately, limited availability is one of the known issues of guide animals due to the lengthy training process and their relatively short working lifespan. Consequently, only a small fraction, approximately two to three percent [2], of potential users are able to benefit from the assistance of guide dogs. Researchers have explored various types of assistive technologies for BVIs, including smart canes [3, 4, 5, 6, 7], belts [8, 9], glasses [10], audio/haptic systems [11, 12], and mobile robots [13, 14, 15, 16, 17, 18]. Recently, the development of affordable and capable quadrupedal robots has opened up opportunities for robot guide dogs [19, 20, 21, 22, 23], which possess notable qualities in terms of autonomy and mobility.

Developing robotic guiding systems for BVIs requires additional challenges beyond the common practices of the existing autonomous robot navigation systems [24, 25]. Unlike conventional formulations that simplify the problem as *Point-goal* or *Object-goal* navigation [26], navigation of the human-robot dyad often involves more complex and subtle bidirectional interactions, including high-level command generation by the user and intelligent disobedience by the robot. Furthermore, ensuring BVI user safety is critical during navigation, yet it is a challenging problem because the sensor configurations are robot-centric.

This paper discusses principles and practical solutions for transforming an autonomous robot into a guide robot for individuals with visual impairments. The topics that we explore include formalizing the wayfinding task of a human and guide dog team, developing a data-driven model of the human and the guide robot interaction, and improving the safety of the BVI user. First, we identify the mechanism of wayfinding in the literature [27, 28, 29] and formally define the guided navigation task using Markov Decision Processes (MDPs). Then we collect real human-robot interaction data of three BVI and six sighted users and develop a concise model, *Delayed Harness*, to better predict the navigation behaviors of the team in simulation. Finally, we introduce an action shielding mechanism [30] to improve the safety of the BVI user, which predicts the human position in the next step and filters out dangerous actions.

A mid-size quadrupedal robot, AlienGo [31], with a rigid harness is chosen as the evaluation platform. Initially, we evaluate the accuracy of the developed *Delayed Harness* interaction model and the improved safety of the learned policy in the Habitat [25] simulator. Then we deploy the developed system on a real AlienGo robot to demonstrate the effective long-range navigation of the human-robot team. Our contributions include: (1) We develop a formal definition of the wayfinding task of the human and guide dog team. (2) We collect human-robot interaction data, develop a concise interaction model, and open-source it to foster guide robot research in the robotics community. (3) We improve the safety of users by employing action shielding. (4) We demonstrate a complete system on hardware that can travel along trajectories for more than 100 meters.

# 2  Related Work

**Assistive Guidance Systems for Visually Impaired People.**  Researchers have developed diverse assistive systems to enhance mobility for visually impaired people. Such assistive systems consist of essential elements, including sensors for perceiving the environment, computers for processing information, and various interfaces to instruct users. One common form is passive hand-held or wearable devices, such as canes [3, 4, 5, 6, 7], belts [8, 9, 11, 12], smart glasses [10], and smartphone applications [32, 33], which generally give audio or haptic feedback to inform users. On the other hand, there exist guide robots [34, 13, 14, 19, 20, 22] that are designed to lead users actively toward destinations. Since the earliest guide robot MELDOG [34], such systems are often built on top of wheeled robots for mobility [13, 14, 15, 35, 16, 17, 36, 37, 38]. Recently, legged robots are emerging in consideration of the ability to travel as real guide dogs [19, 20, 21, 22, 23, 39, 40, 41].

**Human Modeling in Guide Robots.**  Developing guide robots necessitates modeling human-robot interaction because they communicate with users via physical interactions. For instance, Wang et al. [16] introduce a rotating rigid rod model that assumes the user is holding the end of the rotating

rod at a fixed distance. On the other hand, some guide robots [19, 42, 22] communicate via a flexible leash, which investigates a mathematical model for capturing slack and taut modes. Likewise, interactions are typically modeled based on the interface of robots that have been developed [43, 35]. The most common interface for a real guide dog is a fixed harness [44, 45]. In this paper, we claim that a simple offset between the human and the robot is insufficient to capture bidirectional interactions and propose a novel mathematical model to improve the human-robot navigation experience.

**Safety-aware Reinforcement Learning for Navigation.** Autonomous robot navigation has gained significant attention in robotics and has been approached through both planning-based methods [46, 47] and learning-based techniques [48, 49, 50]. Please refer to the survey paper [51, 52] for further detailed references. Numerous algorithms have been developed to enhance the safety of navigation, including action shielding [30, 53], model predictive methods [54], multi-agent strategies [55], and near future considerations [56, 57]. Our work is also inspired by these prior works while being more customized toward the human-robot team navigation task.

## 3 Formalizing the Wayfinding Task

This section identifies how a human interacts with a guide dog and formulates the wayfinding task for a guide robot using Markov Decision Processes (MDPs). Our problem definition aims to develop more natural and comfortable user interfaces of guide robots for BVI users, compared to typical *Point-Goal* or *Objective-Goal* navigation problems [26].

**Wayfinding with a Guide Dog.** We first identify how guide dogs and human handlers communicate to work as a team by reviewing relevant background materials [44, 27, 28, 29] and conducting interviews with actual guide dog users. Typically, the human user is the one who generates high-level directional cues, such as "go straight" or "turn left", based on his or her own knowledge, while the guide dog handles local path following and collision avoidance based on visual perception. Unless the team routinely travels to the same destinations, the guide dog is not typically aware of the goal or path. Then, the guide dog sees and leads the human handler based on the given command. The guide dog needs to follow the virtual "travel line", a centerline of the hallway directed by the handler while adjusting a trajectory to ensure the safety of both the human and the dog itself. We refer to this type of human-dog navigation as *Wayfinding*, and it is our goal to replicate this interaction with a human-robot team.

In addition, a guide dog should understand user intention and compensate for any errors. We find two major types of human error: *Orientation error* and *Timing error*. For instance, *orientation error* can occur in environments such as a long hallway, where the user command is misaligned with the travel line and must be corrected. *Timing error* occurs when a human handler gives turn commands before or after the actual turning point. For instance, if a turning command is given early while traveling down a hallway, the robot is expected to find the next available turn or stop if no turn is available. Please refer to Figure 6 and the supplemental video for illustrative examples.

**Problem Formulation.** We define the wayfinding task for the robot as following high-level human directional cues: *going straight*, *turning left/right*, or *stopping* while adjusting a detailed trajectory to avoid collisions. We use Markov Decision Processes (MDPs) to formalize this wayfinding task, which is defined as a tuple of the state space $\mathcal{S}$, action space $\mathcal{A}$, stochastic transition function $p(\mathbf{s}_{t+1}|\mathbf{s}_t, \mathbf{a}_t)$, reward function $r(\mathbf{s}_t, \mathbf{a}_t)$, and the distribution of initial states $\mathbf{s}_0 \sim \rho_0$. The state and action spaces can vary depending on the choice of the robot. For instance, one common choice is to construct the state using onboard sensors, such as RGB-D or lidar, while defining the action space as a set of possible navigation commands to the robot. We use depth images, lidar readings, and a relative location from the start location as a state and adopt discrete actions based on the recent autonomous navigation papers [49, 58, 50].

The reward function is critical to describe the desirable wayfinding behavior, which is defined as:

$$r_t = (d_t - d_{t-1}) - a\Delta \max\left(|\bar{\theta} - \theta_t| - b, 0\right) - c_t^{\text{collide}} - \lambda \tag{1}$$

where $d_t$ is the human's travel distance from the start point, $\theta$ is the human's orientation, $\bar{\theta}$ is the target orientation (e.g., $0°$ for the straight command and $90°$ for the turn left command), $a$ is a

scalar weight, $b$ is the error margin above which the corresponding term is activated, $c_t^{\text{collide}}$ is a collision penalty, and $\lambda$ is a time penalty. The first term encourages the team to travel as far as possible while minimizing the human's unnecessary movements, such as stepping backward, the second term encourages matching the desired orientation (with some flexibility), and the third and fourth terms regularize unsafe and unnecessary behaviors, respectively.

Because our formulation rewards the team for traveling long distances, it encourages the navigation policy to compensate for any orientation or timing errors, particularly in narrow hallways. Also note that all the terms are defined in a human-centric manner in order to provide a smooth trajectory for the visually impaired user, not for the robot itself.

## 4   Modeling Human-Guide Robot Interactions

In order to develop a successful guiding policy, it is important to simulate human-robot interactions accurately. Guide dogs wear a rigid harness, instead of a loose leash, allowing the human to feel both the guide dog's position and its orientation. We adopt such a fixed harness for our human-robot teams. While there are established human-robot models, such as a rotating rod [16], a slack/taut leash [19], and a geometric model [35], we require a new model specifically tailored to our guide dog robot with a harness. In this section, we propose a new interaction model, the *Delayed Harness*, and optimize the model parameters using real-world data.

**Preliminary: Rigid Harness Model.**   The simplest model for describing rigid harness interactions is to use a fixed offset between the human and the robot. In this model, the human maintains their relative position to the robot, facing the same forward direction. Formally, when the robot position and orientation $(x_t^R, y_t^R, \theta_t^R)$ is given, the human state $(x_t^H, y_t^H, \theta_t^H)$ is computed by assuming a fixed distance $d$ between the handler and the robot as follows:

$$x_t^H = x_t^R + d \cdot \cos \theta_t, \; y_t^H = y_t^R + d \cdot \sin \theta_t, \; \theta_t^H = \theta_t^R. \tag{2}$$

This model expects that the human and the robot rotate together, requiring extra space compared to leash, string, or rod connections.

**Delayed Harness Model.**   A rigid harness model is not accurate because a human cannot exactly follow the robot in reality. As a result, a learned robot policy based on such a model may lead to unnecessary movements or even collisions with a user.

To this end, we develop a more flexible model named *Delayed Harness*. This model is based on the observation that a robot's action causes the change of the relative location, but that a human tends to gradually recover the default relative location (Figure 2). Let us define the robot and human states as $\mathbf{x}_t^H = [x_t^H, y_t^H, \theta_t^H]^T$ and $\mathbf{x}_t^R = [x_t^R, y_t^R, \theta_t^R]^T$, which gives the offset $\mathbf{o}_t = \mathbf{x}_t^H - \mathbf{x}_t^R$. Once the robot takes an action $\mathbf{a}_t$, it first changes its position: $\mathbf{x}_{t+1}^R = \mathbf{x}_t^R + \mathbf{a}_t$. Now, we have a temporary offset $\hat{\mathbf{o}}_{t+1} = \mathbf{x}_t^H - \mathbf{x}_{t+1}^R$ (gray dashed line in Figure 2). We gradually interpolate this temporary offset toward the default offset, $\bar{\mathbf{o}}$ (green line), to obtain the corrected

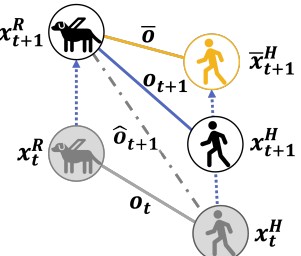

Figure 2: Delayed Harness Model.

offset in the next time: $\mathbf{o}_{t+1} = \alpha \hat{\mathbf{o}} + (1 - \alpha)\bar{\mathbf{o}}$. Finally, we compute the next human position $\mathbf{x}_{t+1}^H = \mathbf{x}_{t+1}^R + \mathbf{o}_{t+1}$. Note that we use simple arithmetic operators for illustration, but in reality, they should be handled as transformation operations.

**Data Collection and Model Fitting.**   Our *Delayed Harness* model requires four parameters: the default offset $\bar{\mathbf{o}} = [\Delta x, \Delta y, \Delta \theta]^T$ and the decaying parameter $\alpha$. We determine these parameters by fitting them to the collected trajectories. We recruit three BVI users and six sighted people, and ask them to walk with a manually-controlled robot guide dog over five trajectories described in the work of Nanavati et al. [35]. The sighted individuals each wore a eye mask while walking with the robot. Please refer to the supplemental material for more details of the data collection process. The collected data is provided as supplemental material (`interaction-data.zip`).

# 5   Improving Safety via Action Shielding

We further introduce an action-shielding approach [30] to improve safety during human-robot navigation. Specifically, our focus is on enhancing the safety of the BVI user based on data from sensors arranged in a robot-centric layout. Inspired by a guide dog user who also uses a cane, our action shielding predicts the changes in the human and robot positions according to the possible actions and filters out unsafe actions. Particularly, we designed the shielding mechanism based on lidar, which has relatively better accuracy ($\pm 5$ cm) than a depth camera (5 % up to 15 m).

The implementation of such action shielding requires two subroutines: (1) predicting possible collisions in the next time step, and (2) suppressing potentially risky actions.

**Computation of Shielding Zone.**   To identify the shielding zone, we compute the occupied areas of both agents in the current and next time steps, which are approximated as circular shapes. While we have four collision primitives, the region between the occupied areas of the human and robot agents should be empty because the two agents are physically connected through a harness. In addition, the zone between the current and the next time

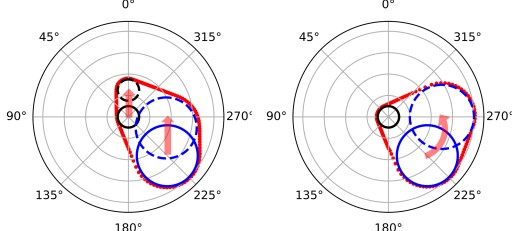

Figure 3: Examples of Shielding Zones.

steps should also be collision-free to account for continuous movements. Therefore, we compute a convex hull of all the collision shapes from both agents at two timesteps to compute the shielding zone (red line in Figure 3). Once computed, we sample rays from the origin to determine the distance thresholds for lidar. Refer to the Appendix 8.2 for more details.

**Suppressing Unsafe Actions.**   When shielding is activated, we can suppress unsafe actions by adjusting the action probability distribution by a *suppression factor* $\beta$. A suppression factor of zero means assigning zero probability to the unsafe actions to force the policy to select safe actions only. It is also possible to employ a small positive value to encourage exploration during training.

The action shielding technique can be utilized during both training and testing. During training, we anticipate the agent will learn a safer policy, especially when sensing is not perfect due to sensor noises or blind spots. Alternatively, we can combine action shielding with any pre-trained policy to improve its safety at the evaluation stage. We examine how the learned policy performs under different action suppression factors in ideal and realistic sensing conditions in Section 6.3.

# 6   Experiments

We conducted experiments to address the following questions: (1) Is the proposed delayed harness model more accurate than other models? (2) How does the interaction model impact performance? (3) Does action shielding effectively decrease collisions in noisy environments? (4) When combined, can the proposed robot guide dog navigate to the destination with a human user?

## 6.1   Training Details

Our policies were trained in the Habitat simulator [25] with the Matterport3D dataset [59] using DD-PPO [49]. The training typically took 10 million steps until convergence, which roughly corresponds to two to three days. The observation space consists of all the sensor information, including depth images, 2D lidar reading, and the relative location from the starting position, followed by a one-hot encoded high-level directional cue provided by the user. For the low-level robot action space, we choose ten discrete actions {stop, forward, turn $10°$ left/right, sidestep left/right, diagonal steps toward the front left/right} inspired by the recent navigation literature [60]. We particularly included side and diagonal steps because they are effective for guiding the user to escape confined regions.

Table 1: Comparison of Different Interaction Models

| Model | Fixed-unopt-ind | Fixed-opt-ind | RR-opt-ind | DH-opt-ind (ours) | DH-opt-all |
|---|---|---|---|---|---|
| RMSE | 136.2 | 102.8 | 431.4 | **86.3** | 123.1 |

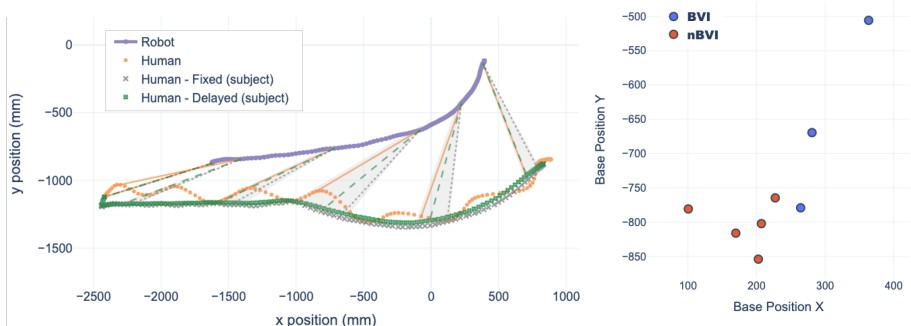

Figure 4: (**Left**) Different trajectories of interaction models optimized using the collected motion capture data on each individual subject. (**Right**) Each subject's base positions derived from the optimized results of the *delayed harness* model.

## 6.2 Comparison of Interaction Models

We first compared our *delayed harness* model against two baselines: a *fixed harness* and a *rotating rod* [16]. By default, the parameters are optimized for each individual subject, which gives us three models: *Fixed-opt-ind*, *RR-opt-ind* (rotating rod), and *DH-opt-ind* (delayed harness). In addition, we included the unoptimized setting, *Fixed-unopt-ind*, which uses the starting parameter to estimate the default offset $\bar{\mathbf{o}}$ and assumes zero decay $\alpha = 0$. We also investigated the *DH-opt-all* model, which optimizes a single set of parameters for all the subjects. For all the optimized models, we fitted the parameters over ten trajectories and validated them over five trajectories. The results indicated that the *DH-opt-ind* model exhibited the highest accuracy, followed by *Fixed-opt-ind*, *DH-opt-all*, *Fixed-unopt-ind*, and *RR-opt-ind* (Table 1). Figure 4 left presents the example of the different trajectories between *Fixed Harness* and *Delayed Harness* models. For the same time step, *Delayed Harness* (connected via green line) is closer to the actual human location (orange line) compared to *Fixed Harness* (grey line) by capturing the delayed response of the human user. It is also worth mentioning that the low accuracy of a rotating rod model was because its interface was different than a harness: therefore, we did not further investigate more complex rod-based models, such as the geometric model proposed by Nanavati et al. [35].

The accuracy difference between *DH-opt-ind* and *DH-opt-all* highlights the importance of models customized for users. We observed that subjects have different default offsets (Fig. 4 right) or responsiveness to direction changes. Therefore, we decided to train policies for each individual, assuming that it is possible to collect user-specific data at onboarding. In the future, it will be interesting to investigate preference optimization techniques [61] to improve robot guide dog behaviors.

Indeed, selecting the appropriate interaction model is critical to achieving the best human-robot team navigation. Table 2 presents the performance of the learned policies using various interaction models at training and testing times. Both policies with the fixed or delayed harness models showed their best performance when evaluated with the corresponding model. However, if the model is altered, the performance drops, approximately 25 % lower than the original performance. Therefore, we suggest to train a policy with a more accurate interaction model, such as *Delayed Harness*.

Table 2: Rewards with Different Training and Testing Interaction Models.

| | | Test | |
|---|---|---|---|
| | | Fixed | Delay |
| Train | Fixed | 5.96 | 4.27 |
| | Delay | 4.00 | 5.04 |

| Sensor Config | Supp. Factor during Train | Supp. Factor during Test | Collision-Free Ep. Ratio ↑ | Avg. Collisions Per Ep. ↓ |
|---|---|---|---|---|
| **Ideal** | 0.0 | 0.0 | (diverges) | |
| | 0.1 | 0.0 | **0.96** | **0.04** |
| | 0.5 | 0.0 | 0.84 | 1.46 |
| | 0.9 | 0.0 | 0.88 | 1.80 |
| | 1.0 | 0.0 | 0.92 | 1.40 |
| (no shielding) | 1.0 | 1.0 | 0.32 | 8.20 |
| **Noisy** | 0.0 | 0.0 | (diverges) | |
| | 0.1 | 0.0 | 0.68 | 4.12 |
| | 0.5 | 0.0 | **0.84** | **1.48** |
| | 0.9 | 0.0 | 0.64 | 2.52 |
| | 1.0 | 0.0 | 0.72 | 3.16 |
| (no shielding) | 1.0 | 1.0 | 0.52 | 3.96 |

Table 3: Performance of Action Shielding in Ideal and Noisy Environments.

## 6.3 Effectiveness of Action Shielding

The main objective of our action shielding mechanism is to improve the safety of the human-robot team in unseen environments with noisy sensor readings. To this end, we compared a policy with action shielding against a vanilla RL policy in both ideal and noisy simulated environments. Additionally, we varied an action suppression factor $\beta$ during training as well, which has a large effect on the performance. For noisy environments, we employed Gaussian noise for the lidar and Redwood [62] noise with an intensity of 1.0 for the depth camera.

In Table 3, we present the collision-free episode ratio and the average number of collisions per episode as metrics for evaluating the safety of the policies. Overall, it is evident that activating action shielding ($\beta < 1.0$) during evaluation effectively reduces the occurrence of collisions. However, the desirable training configurations differed in ideal and noisy environments. In an ideal setting, the performance is not very sensitive to the choice of the suppression factor. It is even very beneficial to turn on action shielding only at the testing stage, which shows a great performance margin against no shielding (collision-free episode ratios: 0.92 vs 0.32). In a noisy environment, training a policy with mild action shielding ($\beta \sim 0.5$) was more effective for adapting a navigation strategy to cope with action shielding. In any configuration, zero action supression $\beta$ during training results in divergence of learning, which is likely due to insufficient exploration.

Action shielding was effective even in real environments with noisy sensor readings. We presented two distinct scenarios where the vanilla policy (without action shielding) failed to generalize to novel situations, such as encountering curtains or large boxes that were unseen during training. Conversely, the policy with action shielding successfully navigated around these obstacles by predicting the user and robot trajectories. For more details, please refer to the supplemental video.

## 6.4 Real-world Experiments

We deployed the proposed robot guide dog system on the hardware of AlienGo [31] by Unitree (Figure 5). This quadruped robot was equipped with a Zed depth camera, a Slamtec RPLIDAR lidar, and an Intel t265 tracking camera. The robot leads the user via a harness while taking verbal commands as directional cues. The policy was trained and deployed with action shielding with action supression of 0.5 and 0.0, respectively, based on Table 3.

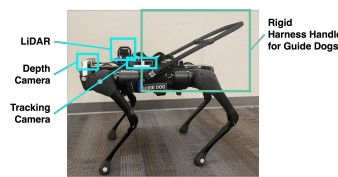

Figure 5: Hardware Platform

**General Navigation.** We assigned two BVI and three sighted users wearing an eye mask the task of completing five indoor routes with four directional cues: `forward`, `left`, `right`, and `stop`. Route A (146 m) is a curved path expected to be traversed by a single forward cue. Routes B (130 m), C (121 m), D (120 m), and E (80 m) required the user to issue multiple turns based on their own localization. Refer to Appendix 8.3 and the supplemental video for the details of the routes. We also set additional obstacles, such as wet floor signs and opened doors, on top of the existing

chairs and trashcans. In our experiments, the user successfully accomplished all the tasks without collisions. In Route A, a single command of `forward` was sufficient to complete all the tasks, even in a very long curved corridor. In Routes B and C, the user sometimes issued turning commands early or late, but the robot was able to compensate for the timing errors. In Routes C, D, and E, a robot intelligently adjusted the trajectory along the hallway without additional cues. Where there were obstacles ahead, the robot guide dog led the user to avoid them while staying in the travel line. Occasionally, the human got close to obstacles or walls, and the robot prevented human collisions by actively using action shielding, which encourages sidestepping as a means of avoidance. After the navigation experiments, we conducted interviews with the BVI users (Appendix 8.3). The users were positive about the system's controllability and found their experience improved with familiarity during the second trial.

**Orientation and Timing Errors.** As explained in Section 3, the guide robot must effectively handle cues that are provided with some timing or orientation errors caused by human users. We showcase the robustness of the developed robot guide dog by displaying the trajectories involving human errors. Figure 6(a) displays *orientation errors* on a forward cue, where the robot corrected the initial orientation error and followed the direction of the hallway to travel as long as possible. Figure 6(b) illustrates the turning with *timing errors*, where the

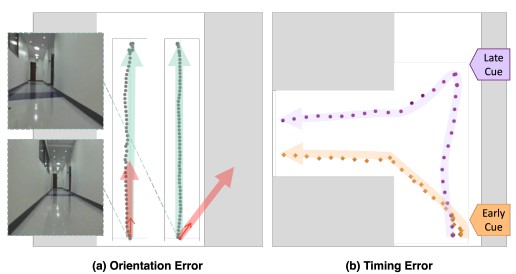

Figure 6: Trajectories with Two Type of Human Errors: Orientation Error (a) and Timing Error (b).

robot took diagonal stepping or 120° turning to correct early or late issued commands, respectively.

## 7   Conclusion and Limitation

This paper presents three topics and potential solutions for transforming a standard quadrupedal robot into a robot guide dog to assist blind or visually impaired individuals. First, we study the navigation patterns of human-guide dog pairs and establish a formal wayfinding task using Markov Decision Processes (MDPs). Then we propose a concise interaction model called the *Delayed Harness* to effectively represent the interaction between a human and a robot guide dog, which leads to more accurate behavior prediction and better navigation performance. The parameters of this interaction model are optimized with respect to the real interaction data of three visually impaired and six sighted subjects, which is provided as supplemental material to this manuscript. Finally, we introduce an action shielding mechanism to improve the safety of the human user, inspired by a guide dog user who also uses a cane along with a guide dog. We demonstrate that the integrated robot guide dog can navigate with users over multiple 100+m long trajectories. We hope that the topics and techniques discussed in this study will inspire further research in the development of guide robots for blind and visually impaired individuals.

**Limitation.**   The proposed solutions in the paper have room for improvement, and we plan to investigate the following research directions. Firstly, while we have explored only a limited set of basic verbal cues, real guide dog users employ a wider range of commands, such as "find an elevator" or "follow the person", combined with non-verbal cues. It will be beneficial to formalize a more expanded set of human-guide dog interactions. Additionally, although our *Delayed Harness* model effectively captures typical navigation behaviors, it struggles with out-of-distribution situations like users pausing to chat with someone. In the future, we will expand the provided dataset to encompass such diverse scenarios. The developed action shielding mechanism is effective but not perfect. For instance, it can fail even in simulation due to blind spots of the lidar sensor. To resolve this issue, we will explore alternative sensors and safety mechanisms to improve the safety of the human and the robot. Our proposed system requires an hour-long data collection session for personalized navigation. To mitigate this, we will explore online adaptation algorithms to reduce training time. Finally, we plan to evaluate the system on a larger population of blind or visually impaired users to gather feedback for further improvements.

**Acknowledgments**

This research was supported by Google Research Collabs and Google Cloud. JK was supported by the National Science Foundation Graduate Research Fellowship under Grant No. DGE-2039655. Any opinions, findings, and conclusions or recommendations expressed in this material are those of the author(s) and do not necessarily reflect the views of the National Science Foundation, or any sponsor.

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

# 8 Supplementary Material

## 8.1 Human-Robot Modeling Data Collection

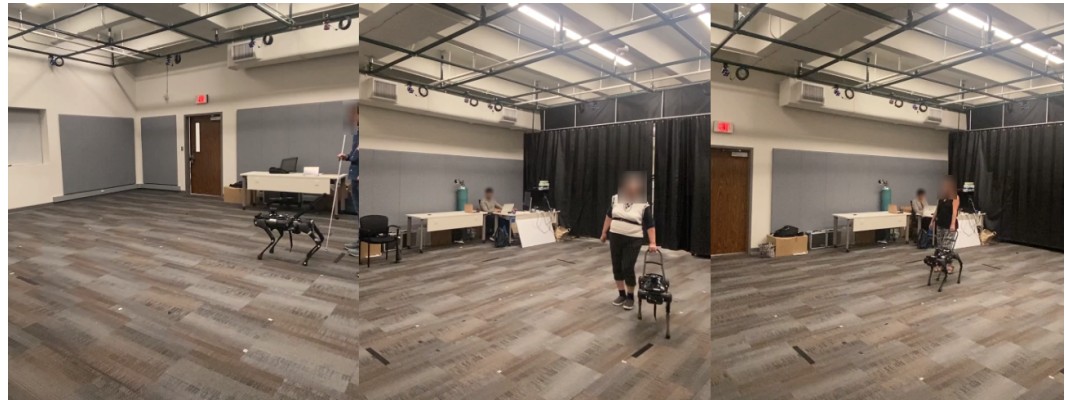

Figure 7: Data collection of three blind or visually impaired (BVI) subjects.

We collected human and robot interaction data using a Vicon motion capture system with Pulsar active marker clusters (Figure 7). The study is reviewed and approved by the institutional review board (IRB). We recruited nine participants consisting of three blind or visually impaired (BVI) and six sighted subjects. For the data collection, the participants are asked to wear a back strap to place a motion capture marker on their lower back. Then, they follow the robot, AlienGo, by using their left hand to hold onto a harness handle that is attached to the robot. For white cane users, we asked them to hold their cane in their right hand for safety purposes, but they did not actively use the cane during data collection.

Table 4: Description of Five Robot Trajectories for Interaction Data Collection

| Trajectory | Description |
|---|---|
| 1 | 2.5 m forward, 90-degree in-place left turn |
| 2 | 1.2 m forward, 90-degree left turn |
| 3 | 0.75 m forward, 45-degree gradual left turn, 135-degree in-place right turn |
| 4 | 0.5 m forward, 90-degree right turn, 0.5 m forward, 90-degree left turn, 0.5 m forward |
| 5 | 0.6 m forward, 90-degree in-place right turn, 0.6 m forward, 180-degree left u-turn, 0.6 m forward, 90-degree right turn, 0.6 m forward |

**Robot Trajectories.** The robot is scripted to follow pre-defined trajectories as described in Table 4. The five trajectories are based on the work of Nanavati et al. [35] with modifications to fit in the motion capture space.

**Experimental Protocol.** The data collection process is divided into three parts. Each part is designed to examine different aspects of the interaction behavior. *Part 1:* The subjects followed the robot without vision, but relying purely on the physical interaction without any visual cues. We repeated each trajectory three times in random order. To prevent the subject from predicting the trajectory, they were told that there were 15 trajectories. *Part 2:* The subjects followed the robot without vision, but with prior information on the robot's expected movement. We provided verbal descriptions of the expected cue just before the robot executed each cue. *Part 3:* The sighted subjects followed the robot with their full vision. This is extra data collected only from sighted subjects to respond to the robot's movement with additional visual perception.

## 8.2 Pseudocode of Action Shielding

This section provides a more detailed algorithm of our action shielding mechanism. We compute the convex hull of the collision primitives first, compute the lidar thresholds from this hull, and then check whether the action is safe or not. Please review this material with Section 5.

---

**Algorithm 1** Convex-hull Action Shielding.

---

**Input** current position and orientation of robot and human $\mathbf{x}_t^R$, $\mathbf{x}_t^H$, a list of possible actions $\mathbf{A}$, a vector of action probability $\mathbf{p}$, action suppression scale $\beta$, lidar reading $\mathbf{l}$.
**Output** A modified action probability vector $\hat{\mathbf{p}}$

1: **procedure** ACTIONSHIELDING($\mathbf{T}_r^t$, $\mathbf{T}_h^t$, $A$, $s$, $l$)
2:     **for** $i \leftarrow 1$ to len($\mathbf{A}$) **do**
3:         $a = \mathbf{A}[i]$
4:         $\hat{\mathbf{p}}[i] = \mathbf{p}[i]$
5:         $\mathbf{x}_{t+1}^R, \mathbf{x}_{t+1}^H \leftarrow \text{EstimateNext}(\mathbf{x}_t^R, \mathbf{x}_t^H, a)$          ▷ Estimate Via Interaction Model
6:         $\mathcal{S} \leftarrow \text{ConvexHull}(\mathbf{x}_t^R, \mathbf{x}_t^H, \mathbf{x}_{t+1}^R, \mathbf{x}_{t+1}^H)$
7:         $\bar{\mathbf{l}} \leftarrow \text{ComputeLidarThresholds}(\mathcal{S})$
8:         **if** any of reading $\mathbf{l}$ is less than the threshold $\bar{\mathbf{l}}$ **then**
9:             $\hat{\mathbf{p}}[i] \mathrel{*}= \beta$
10:         **end if**
11:     **end for**
12:     $\hat{\mathbf{p}} \leftarrow \text{normalize}(\hat{\mathbf{p}})$
13:     **return** the modified action probability $\hat{\mathbf{p}}$
14: **end procedure**

---

## 8.3 Real World Navigation Experiments

**User Study**     Overall, the users were positive about our suggested system: *"I think the technology is going to be great. Once you tell it to go straight, it's in a straight path, and that's very nice. ... I like the way it was pulling– it was very consistent, a nice solid pull forward."* The users enjoyed the developed system's controllability, allowing the users to navigate the hallway effectively: *"I like being able to do the commands. ... So that was good, a little more control over the direction and the way the dog (robot) was walking."* The users also felt much more comfortable with the second trial than the first one because they became familiar with the system: *"I thought the second run was like 50 % better than the first one."* However, one participant suggested reducing the noise from the gait: *"(Q: What did you like least about this experience?) Well, the noise from the gait."*

**Teamed Navigation Routes**     We include here the map information for our real-world experiments (Figure 8 and Figure 9).

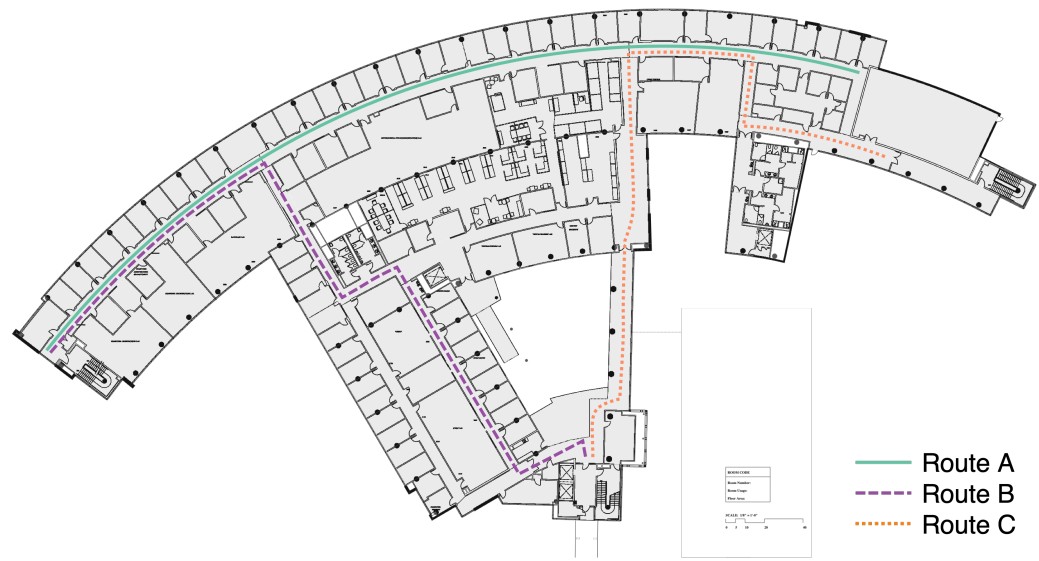

Figure 8: Illustration of three routes in our experiments.

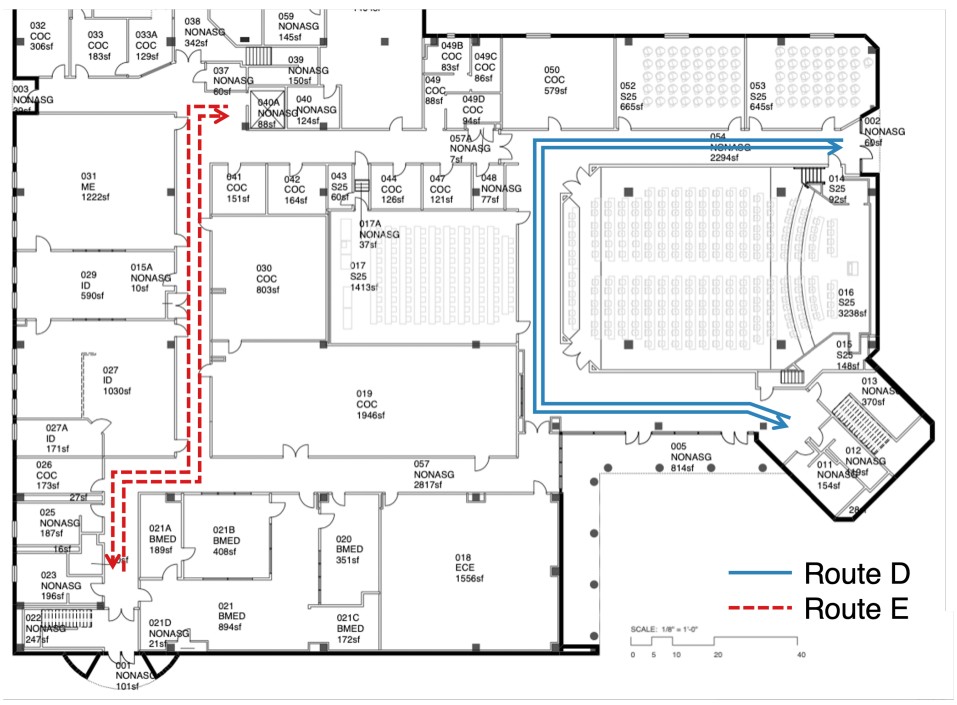

Figure 9: Illustration of two routes in our experiments with the visually impaired subjects.

