# OpenReview forum: "Transforming a Quadruped into a Guide Robot for the Visually Impaired: Formalizing Wayfinding, Interaction Modeling, and Safety Mechanism"
_robot-learning.org/CoRL/2023/Conference — CoRL 2023 Poster_

### Official Review · Reviewer_dd3b · 2023-07-19

**Confidence:** 4
**Originality:** Very Good
**Technical Quality:** Excellent
**Clarity Of Presentation:** Good
**Impact:** 4

**Recommendation:**

Strong Accept: I recommend accepting the paper and will argue for my recommendation even if other reviewers hold a different opinion.

**Review:**


### Strengths

- **S.1** The paper is very focused and in my opinion sets a very reasonable goal (model human-robot interactions better) and goes above and beyond accomplishing this goal (by not just recording their own data but also looping back and deploying the method on a real robot).
- **S.2** The writing is clear and concise.
- **S.3** As far as I can tell, the method is novel and straight-forward (as in "easy to follow").

I also like that you're re-stating the research questions at the beginning of the experiment section.
I also massively appreciate the release of the interaction dataset with the submission of the paper. This makes everything more reproducible.

### Weaknesses

There aren't any major downsides to this for me and the only things I'm criticizing are lack of polish.

- **W.1** The figures could use some work. I'd include a better caption for Fig.2 that actually summarizes the model, also in Fig. 2, the corrected offset variables are in the center overlapping the lines and hard to read. In general, maybe there's a better way to represent this in terms of colors and maybe it's worthwhile to replace the anonymous geometric shaped with a stylized robot dog and human. Fig.1 is a neat header graphics that shows what you're doing but the right side doesn't actually show action shielding - you could draw over the photos (like arrows on the floor) that indicate how action shielding prevents a bad action in the situation. In Fig.3, the fonts are too small. Fig. 4 (left) is really hard to parse for me. In Fig. 4 (right), I don't really know what to do with that information.
- **W.2** Is there any way to quantify the results of the real-world experiments or are they more of a demo? How are the cues given?
- **W.3** I don't fully understand why the action shielding is only applied during training and at test time, it's binary on-or-off (cf. Tab.3).
- **W.4** In your limitations, you mention gaps in the lidar but you're not controlling the robot at the same framerate that the lidar is outputting data, right? So why not accumulate the lidar data?


**Quality Of The Limitations Section:**

Limitations are addressed clearly

**Questions For Rebuttal:**

See **W.2** and **W.3** above.
Also,

- **Q.1** How does the human indicate their desired trajectory? In the video, it looks like waving in the general direction.
- **Q.2** I get how this method helps with turn indications that come too early (due to interpolation) but I don't fully understand why that also works when the turn indication is too late.
- **Q.3** Any idea why PPO diverges in some cases (Tab.3)?
- **Q.4** I don't fully understand how PPO is trained or even set up in Habitat - You model both the human and dog there, right? I assume the dog is taking a random path through the house, based on a randomly sampled goal? How do you deal with the user's direction indications? How do you terminate an episode?

**Robotics Focus:**

Sufficient demonstration on hardware

**Summary Of Paper:**

The paper works on getting robot dogs to be better guide dogs for the blind and visually impaired (BVI).
The authors introduce a new formalism for modelling guide-dog-following, they collect their own dataset an verify that this formulation indeed models the data. They also introduce a new action filtering system dubbed "action shielding", that reduces the number of collisions that the BVI-dog pair encounters and they evaluate that in sim. Lastly, they also deploy their policy and action filtering method to a real robot and verify qualitatively that everything works.

**Summary Of Recommendation:**

This paper is an easy recommendation for me. It's set out to do a seemingly simple thing, shows that current approaches don't work as well, and then goes above and beyond (custom dataset, sim experiments, real-robot experiments) to show the merits of its method.

---

### Official Review · Reviewer_SGRs · 2023-07-19

**Confidence:** 4
**Originality:** Good
**Technical Quality:** Fair
**Clarity Of Presentation:** Very Good
**Impact:** 3

**Recommendation:**

Weak Reject: I recommend rejecting the paper, but will not argue for my recommendation if the majority of other reviewers have a different opinion.

**Review:**

Strengths:
- The authors conduct relatively comprehensive testing of the system in simulation
- Some blind/low-vision participants were included in the data collection (see below for remaining concerns).
- The proposed approach is a potentially promising starting point, although more research is needed to confirm this.


Weaknesses:
Overall, the validation does not provide strong support for the efficacy of the proposed method.  More appropriate testing is needed to confirm that this approach is a step in the right direction.

Most importantly, technologies for BLV users cannot be validated by blindfolding sighted users (see citations below), both for ethical reasons and because there is no reason to think that a blindfolded sighted person moves through the world in the same way as someone who is blind or low-vision.  This is easy to see in Figure 4 in the authors' own work, where there is a clear separation between the BLV participants' behavior and the sighted participants' behavior with the robot.  Disability simulations are to be avoided, but if they are used, it is imperative that the authors are extremely careful about their claims, which is not the case here -- for example, it is not true that the method "improve[s] the safety of the BVI user", since the system isn't actually tested with blind participants.  The authors must have some connection to the BLV community, since their initial data collection included BLV participants; validation should also be conducted with this community.

- I would like to see more intuition for whether the simulation results are meaningful.  How can we interpret the error in the models? Is a difference of <20 in the RMSE meaningful to a user?

- For most values of the suppression factor the average collisions per episode is greater than 1 even with perfect sensing, and 2-4 in many cases with noisy sensors.  This seems high for the context of guiding a blind person. Even 92% collision-free seems low for an action shielding method that is intended to prevent unsafe actions entirely.

- Testing is conducted in an indoor hallway without obstacles; it is unclear why a legged robot is needed in this context.  The paper would be improved by the inclusion of some testing with obstacles for which a legged robot would be useful (e.g., stairs or uneven surfaces).  It is unclear whether the proposed method would work in such a context.

- The validation in 6.4 is conducted with only a single participant, with no information about the recruitment of that participant.  Was this participant a member of the study team and/or a robotics expert? Was the validation study approved by a university ethics board?

- The following work should probably be cited:
S. Azenkot, C. Feng and M. Cakmak, "Enabling building service robots to guide blind people a participatory design approach," 2016 11th ACM/IEEE International Conference on Human-Robot Interaction (HRI), Christchurch, New Zealand, 2016, pp. 3-10, doi: 10.1109/HRI.2016.7451727.

Citations on disability simulation:

French, S. (1992). Simulation exercises in disability awareness training: A critique.
Disability, Handicap & Society, 7, 257–266 10.1080/02674649266780261.
doi:10.1080/02674649266780261

Garreth W. Tigwell. 2021. Nuanced Perspectives Toward Disability Simulations from Digital Designers, Blind, Low Vision, and Color Blind People. In Proceedings of the 2021 CHI Conference on Human Factors in Computing Systems (CHI '21). Association for Computing Machinery, New York, NY, USA, Article 378, 1–15. https://doi.org/10.1145/3411764.3445620

Leo, J., & Goodwin, D. (2016). Simulating Others’ Realities: Insiders Reflect on Disability Simulations, Adapted Physical Activity Quarterly, 33(2), 156-175. Retrieved May 22, 2023, from https://doi.org/10.1123/APAQ.2015-0031

Nario-Redmond, M. R., Gospodinov, D., & Cobb, A. (2017). Crip for a day: The unintended negative consequences of disability simulations. Rehabilitation Psychology, 62(3), 324–333. https://doi.org/10.1037/rep0000127

Cynthia L. Bennett and Daniela K. Rosner. 2019. The Promise of Empathy: Design, Disability, and Knowing the "Other". In Proceedings of the 2019 CHI Conference on Human Factors in Computing Systems (CHI '19). Association for Computing Machinery, New York, NY, USA, Paper 298, 1–13. https://doi.org/10.1145/3290605.3300528

Jonathan Lazar, Jinjuan Heidi Feng, Harry Hochheiser, "Chapter 16 - Working with research participants with disabilities", Editor(s): Jonathan Lazar, Jinjuan Heidi Feng, Harry Hochheiser, Research Methods in Human Computer Interaction (Second Edition), Morgan Kaufmann, 2017, Pages 493-522, ISBN 9780128053904

**Quality Of The Limitations Section:**

Additional details required

**Questions For Rebuttal:**

How would it change the work not to use the disability simulation? Which results can be justified without recourse to the assumption that a blindfolded sighted person is a good model for a blind person? Which results are called into question? For this reviewer, the authors would need to craft a compelling argument that the results stand if a blindfolded sighted person is a poor model of a blind person.  Please also address the handful of questions above.

**Robotics Focus:**

Sufficient demonstration on hardware

**Summary Of Paper:**

This paper presents an RL-based method for controlling the motion of a quadriped robot that is intended for use as a guide for blind and low-vision (BLV) people.  The main contributions are a model of the relationship between the user and robot that accounts for delays in human following and an action shielding approach intended to prevent unsafe actions.

**Summary Of Recommendation:**

Overall, there are potentially interesting elements to the method, but given that the main innovation is that the system is intended to be useful as a guide for blind and low vision users, there is not sufficient evidence that the system is fit for this proposed purpose.  I encourage the authors to conduct a proper validation with at least 5-10 blind and low-vision participants and appropriate baselines and resubmit to a future venue.

Post-rebuttal comments: I think the authors have addressed some of my key ethical concerns (although I am a bit dubious about the use case where a sighted person is in fog or smoke in an emergency situation; in this case, the ecological validity problems are with the scenario rather than the users), so I raise my score to a weak reject rather than a strong reject.  As I said before, I encourage the authors to conduct a more strongly valid and appropriately-powered evaluation and resubmit to a future venue.

---

### Official Review · Reviewer_1S13 · 2023-07-20

**Confidence:** 3
**Originality:** Good
**Technical Quality:** Good
**Clarity Of Presentation:** Good
**Impact:** 2

**Recommendation:**

Weak Accept: I recommend accepting the paper, but will not argue for my recommendation if the majority of other reviewers have a different opinion.

**Review:**

Strengths:
- In general, I found the paper to be well written, easy to follow, and enjoyable to read
- The problem setting is well motivated and clearly described
- The proposed problem formulation and approach seem well designed
- The empirical results presented in the paper are promising

Scope for improvement:
- The assumption that it is possible to collect user-specific data at onboarding seems like a big area for improvement. I would like to see at least more discussion about the limitation and possible directions for a more generalizable/scalable approach.
- There was a very limited number of participants used for model fitting (3 BVI users and 6 sighted users).  I think the paper would be stronger with a much larger and more diverse set of users.  I also question the over-representation of sighted users — as I would expect the behavior of BVI users vs sighted users (even blindfolded) to be different.
- I would like to see more details about the 10 training trajectories and 5 evaluation trajectories (e.g. how were the trajectories chosen and how do they vary in length and difficulty? Do you have reason to believe your methods will generalize/scale beyond the 5 evaluation trajectories?)
- The reported results of Action Shielding seem very promising. However, I would like to see more details about the experiments (similar to the last section — e.g. how many environments are being used for train/test and how do they vary? How are trajectories synthesized in simulation? How is the human state synthesized in simulation?)
- The real-world experiments seem very limited. If I’m understanding correctly, only a single sighted user participated and the system was never evaluated with any BVI users. This seems like a big area of improvement to me. As BVI users are used to motivate the paper, I think the paper would be much stronger with a more rigorous evaluation that includes more BVI participants.

**Quality Of The Limitations Section:**

Additional details required

**Questions For Rebuttal:**

- Is the use of blind-folded sighted users common in training/evaluating systems intended for BVI users? Is there some evidence that shows that blind-folded sighted users behavior is a sufficient emulation of BVI users behavior? My assumption is that the behavior of BVI users vs blind-folded sighted users would be different.  I don't think it's necessarily a problem that you include samples from blind-folded sighted users, but I do question the over-representation of sighted users in your model fitting (and that your subsequent on-robot evaluation is done with a sighted user).  Given that the paper is motivated by BVI users, I think the paper would be much stronger with a more rigorous evaluation that includes more BVI participants.  At the very least I would like to see this limitation addressed with some discussion about your decision to use sighted blind-folded users (and ideally some evidence to show that is sufficient).
- I would like to see more details about the 10 training trajectories and 5 evaluation trajectories (e.g. how were the trajectories chosen and how do they vary in length and difficulty? Do you have reason to believe your methods will generalize/scale beyond the 5 evaluation trajectories?)
- You might consider being more explicit about what ‘accuracy’ is measuring in the interaction model comparison. From the model described in Section 4 and the plot in Figure 4, it seems like this is measuring accuracy at predicting the human state based on the robot position and orientation. But you might consider making that more clear.
- The reported results of Action Shielding seem very promising. However, I would like to see more details about the experiments (similar to the last section — e.g. how many environments are being used for train/test and how do they vary? How are trajectories synthesized in simulation? How is the human state synthesized in simulation?)

**Robotics Focus:**

Sufficient demonstration on hardware

**Summary Of Paper:**

This paper explores the application of a quadrupedal robot as a replacement for guide dogs. The authors formalize the wayfinding task (wherein a human interacts with a guide by giving discrete navigation commands) as an MDP.  To model the human-guide interaction the authors propose a ‘Delayed Harness’ model that accounts for the human’s inability to rigidly follow the guide’s harness without a delay. Additionally, the authors introduce an action-shielding model designed to improve safety during navigation. The model predicts the changes in the human and robot positions and attempts to filter out unsafe actions.

To evaluate their proposed approaches, the authors first conduct a series of experiments situated in Matterport3D environments using the Habitat simulator.  Finally, the proposed system is deployed on an AlienGo robot for evaluation in the real world.

**Summary Of Recommendation:**

I would have liked to see a much more rigorous evaluation that includes more BVI participants (as BVI users are central to the motivation of the paper). In lieu of that, I would like to at least see some discussion of this limitation and some argument for why blind-folded sighted user data is sufficient.  Otherwise, I am left questioning whether these methods scale to the types of navigation interactions you might encounter "in the wild" with real BVI users.

However, I do think the problem is well motivated, the proposed formulation and approach are sensible, and the empirical results presented in the paper are promising. For these reasons, I think the paper is interesting and worth sharing with the community. So I am weakly recommending "Accept".

---

### Author Response · Authors · 2023-08-15
**We would appreciate more comments**

Dear reviewers and AC,

We posted the rebuttal a few days ago. The end of the author-reviewer discussion is approaching, and we would love to hear more feedback. At this point, the biggest concern seems to be the lack of evaluation with BVI users. We tried to address the concern as much as possible, which included one additional study with a BVI user and two more scheduled ones. Please refer to the comments below for more detailed arguments. And let us know if you have any other questions or concerns.

---

### Decision · Program_Chairs · 2023-08-30

**Decision:**

Accept (Poster)

**Comment:**

The reviewers commend the paper for being well-written and organized. The formalization of quadrupedal robots using RL to assist as guide dogs for those with visual impairments presents some interesting ideas. Reviewer SGRs notes several areas for paper improvement, and I would encourage the authors to focus additional efforts on addressing remaining points raised in the reviews.